# Workshop-based participatory research design through the lens of a culture-centered approach (CCA)

Modi Al-Moteri[1]*, Raneem Mohammed Althobiti[1], Rahaf Talal Alsalmi[1], Nadeen Ibraheem Almalki[1], Waad Moala Alsufyani[1], Ghida Abdullah Alghamdi[1], Shahad Fahad Albogami[1], Mjd Fahid Alotaibe[1], Nora Naeem Alameen[1], Jamil Aljuaid[2], Maaidah M. Algamdi[3], Saud Owaydh Almutiry[4]

1 Medical Surgical Nursing Department, College of Nursing, Taif University, POB, Taif, Saudi Arabia, 2 Children's Hospital. At Taif, Saudi Arabia, 3 Community and Psychiatric Health Nursing Department, Faculty of Nursing, University of Tabuk, 4 Al Rass General Hospital, Al-Qassim Health Cluster, MOH, Al Rass, Saudi Arabia

* m.motairy@tu.edu.sa

## Abstract

### Purpose

Nurses in the neonatal intensive care unit (NICU) play a crucial role in health communication, yet their voices are often overlooked. This study explores communication challenges from nurses' perspectives to develop a sustainable communication infrastructure.

### Methods

A workshop-based participatory design (WBPD) was used, where NICU nurses worked collaboratively to identify communication barriers. Thematic analysis was conducted using the culture-centered approach (CCA) to ensure context-specific solutions, integrating both researcher insights and NICU-based communication infrastructure design.

### Findings

A multilevel communication infrastructure model was developed to enhance nurse-parent interactions in the NICU. Key findings highlight that effective communication hinges on three interconnected factors: (1) nurses' skills and access to resources, (2) institutional policies supporting standardized protocols and mentorship programs, and (3) systemic mechanisms for fostering shared understanding. Participants advocated for structured training, culturally responsive practices, and language support tools to address diverse needs. The proposed model integrates

Data availability statement: All relevant data are within the manuscript.

Funding: This research was funded by Taif University, Saudi Arabia, Project No. (TU-DSPP-2024-282). The funders had no role in study design, data collection and analysis, decision to publish, or preparation of the manuscript.

Competing interests: NO authors have competing interests.

learner-centered training, interprofessional collaboration, communicative algorithms, and healthy boundaries to establish a cohesive, inclusive framework.

## Conclusions

Nurse-led, multilevel interventions are essential for improving NICU communication. The proposed model enhances training, policies, and culturally responsive strategies, supporting more effective nurse-parent interactions and improved neonatal care. Beyond the NICU, this framework offers a transferable model for enhancing communication in other high-stress healthcare environments, ensuring more inclusive and structured communication practices across diverse settings.

---

## Introduction

The experiences of parents with newborns in the neonatal intensive care unit (NICU) are often emotionally challenging and stressful [1]. Effective communication between NICU nurses and parents is essential for enhancing parental understanding, reducing anxiety, and improving overall satisfaction [2]. Strong communication also contributes to better patient outcomes by promoting parental involvement in care, improving adherence to medical instructions, and reducing stress-related complications in newborns [3]. Studies show that when nurses effectively communicate with parents, there is greater trust in medical decisions, improved emotional well-being, and enhanced coordination of neonatal care, leading to better health outcomes for both infants and caregivers [4]. Despite this, studies continue to report significant challenges in NICU communication, leading to negative parental experiences [3,5].

Research highlights multiple barriers to effective communication, including the absence of standardized NICU communication guidelines [6], language barriers between nurses and caregivers [7], excessive workload demands on NICU staff [6], and inadequate work conditions for nurses. These factors indicate that communication failures are not solely due to individual shortcomings but rather result from broader structural and organizational constraints [8].

Traditional health communication approaches have primarily focused on message delivery, assuming that clear messaging alone ensures effective communication [9,10,11]. However, this approach does not adequately address the systemic barriers, workplace policies, and organizational culture that shape provider-parent interactions [12]. Notably, the voices of NICU nurses—who are central to parent-provider communication—are often overlooked in research and policy discussions [3]. This study aims to bridge this gap by investigating how NICU nurses perceive communication challenges and identifying systemic improvements to create a more effective communication infrastructure in NICUs. By shifting the focus from individual nurse competencies to systemic, nurse-driven solutions, this study highlights actionable strategies for integrating nurses' perspectives into institutional policies and training

programs. The findings aim to inform NICU staff training programs, institutional communication policies, and broader healthcare frameworks to improve both patient outcomes and provider well-being.

## Methods

A workshop-based participatory research design (WBPD) was employed. A participatory workshop is a collaborative scientific research method that encompasses an active contribution of stakeholders with researchers to address research questions and propose solutions within a community [13,14]. For the current study, a participatory workshop was used as it describes the intentional sharing relationships between stakeholders—a community of NICU nurses and the study research team [15]. In this workshop, the NICU nurses participated as active stakeholders, sharing their lived experiences, identifying communication barriers, and co-developing solutions. They were key participants whose insights shaped the findings and recommendations of the study. The research team acted as active facilitators, facilitating the discussions and ensuring that nurses' perspectives were central to the process. According to Amauchi et al. [13], the deeper the participation, the greater the opportunity for participants to enthusiastically contribute to the development of the expected solutions set out for the research.

### Ethical approval

Ethical approval was granted on the 3rd of June 2024 from the Ministry of Health review board (IRB number: H-02-T-123).

### Culture-centered approach

The culture-centered approach (CCA) serves as the analytic framework for this study. Developed by Dutta [17], the CCA is a theoretical framework in health communication that challenges top-down models by advocating for marginalized voices in the design of health interventions. It emphasizes that effective communication emerges from engaging with the lived experiences of the affected communities. Rooted in the principles of participatory design, the CCA emphasizes co-creating communication infrastructures by centering the voices of NICU nurses, enabling their active participation in decision-making, and fostering grassroots solutions [16]. By prioritizing nurses' perspectives, the CCA empowers them to design interventions that address systemic inequities in health outcomes [17]. The CCA examines health communication through three interconnected dimensions—structure, culture, and agency (Fig 1)—with communication emerging at their intersection [17].

- **Culture** encompasses community values, beliefs, and practices related to health and illness.

- **Agency** refers to individuals' capacity to make choices and take action to address challenges.

- **Structure** represents institutional guidelines, policies, and norms that enable or constrain health behaviors.

This framework dynamically interprets lived experiences, making it uniquely suited to explore NICU nurses' engagement in health communication and identify systemic barriers to effective practice.

### Procedure

**Participant recruitment.** A purposive sampling technique was employed to recruit participants based on their knowledge, experience, and interest in NICU communication challenges. Selection criteria included identifying individuals who were knowledgeable about the topic, contributed meaningful insights, held influence over communication practices, or were directly impacted by potential improvements. A total of thirty-six NICU nurses were identified as eligible and were formally invited via email by the primary researcher on June 5th, 2024. The invitation outlined the study's objectives, participation expectations, and the workshop schedule. Of those invited, twelve NICU nurses expressed interest and

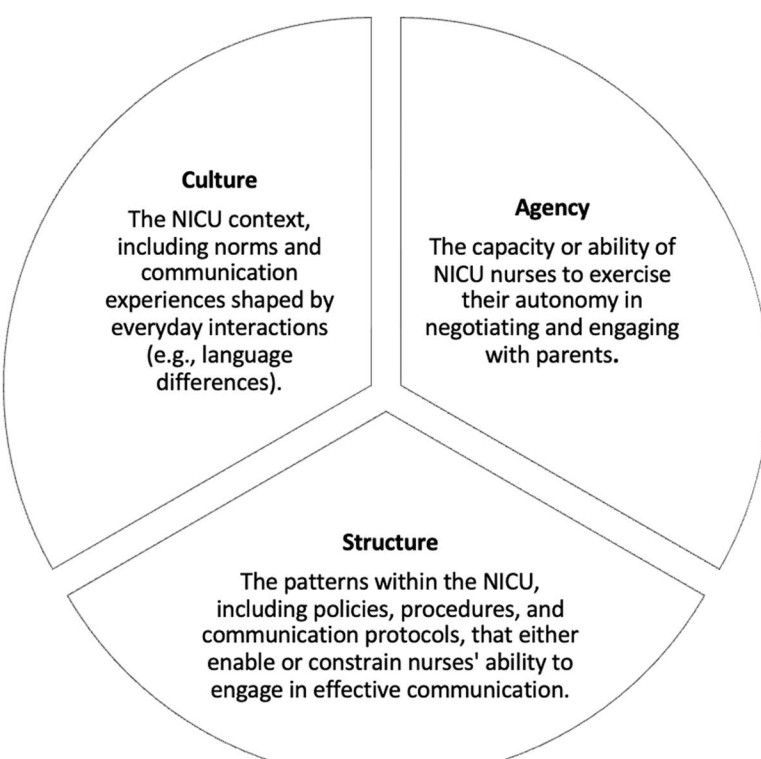

**Culture**

The NICU context, including norms and communication experiences shaped by everyday interactions (e.g., language differences).

**Agency**

The capacity or ability of NICU nurses to exercise their autonomy in negotiating and engaging with parents.

**Structure**

The patterns within the NICU, including policies, procedures, and communication protocols, that either enable or constrain nurses' ability to engage in effective communication.

**Fig 1. CCA theoretical framework.**

responded affirmatively. However, on the day of the WBPD, three participants withdrew due to work-related commitments, leaving nine NICU nurses who provided written informed consent and participated in the workshop.

**Workshop organization and sessions.** The WBPD was conducted from June 10th to 16th, 2024, in a closed meeting room within the hospital library. The space was equipped with materials to foster engagement, including sticky notes, dry-erase boards, flip charts, and colored markers. The workshop aimed to (1) identify communication challenges in the NICU, (2) analyze barriers thematically, and (3) co-design a communication infrastructure to address them. Sessions balanced facilitator-guided discussions with participant-driven insights.

First sessions— On June 10th, 2024, the first session opened with a 20-minute icebreaker where nine nurses introduced themselves and shared their motivations for joining the workshop. Participants then divided into two groups, each with NICU nurses and a researcher facilitator. After outlining WBPD principles (trust, ethics, cultural sensitivity) and securing consent, the team launched a 60-minute structured discussion. Nurses collaboratively mapped communication challenges using sticky notes, flip charts, and dry-erase boards to categorize ideas and visualize connections, while colored markers prioritized ideas. This multimodal approach transformed abstract issues into concrete, actionable insights. Cross-group interactions ensured dynamic dialogue, with facilitators prompting participants to refine solutions and share perspectives between teams.

Second session— From June 11th to 15th, 2024, the research team analyzed transcripts, sticky notes, and facilitator notes to identify recurring themes, such as structural constraints (e.g., lack of protocols) and cultural gaps (e.g., language disparities).

**Third and Fourth sessions**—On June 16th, 2024, two sequential sessions were conducted to finalize the study's outcomes:

The Validation session (11:00 AM–12:00 PM, 60 minutes) involved six participants reviewing and refining themes to ensure alignment with their lived experiences. During this session, the researcher led the discussion while nurses actively validated the themes based on their practical knowledge and experiences.

The Final Co-Design session (1:00–2:00 PM, 60 minutes) engaged the same six nurses in co-designing a communication infrastructure model. Unlike the previous session, this phase maintained a fully participatory approach, where both researchers and NICU nurses were equally involved in co-design activities. The themes validated in the previous session served as a foundation for guiding the development of the model, ensuring alignment with nurses' lived experiences and communication needs. Researchers and NICU nurses integrated frontline experiences with theoretical insights, ensuring the model was practical and evidence-based. During this session, participants proposed actionable strategies to improve the communication process, focusing on standardizing protocols, enhancing interdisciplinary collaboration, and implementing culturally responsive communication.

## Data analysis

A deductive thematic analysis of the collected data was conducted using Braun and Clarke's six-step framework [18]. The first step of the analysis involved data immersion, where eight research team members engaged in repeated readings of the transcripts to develop a deep understanding of the content. Once familiarized with the data, the research team applied a deductive coding approach based on the CCA framework, categorizing data under structure, culture, and agency. Additionally, an open coding process was employed to capture new themes that emerged from participants' responses. Each transcript was coded manually and independently by at least two researchers to enhance inter-rater reliability and minimize individual biases during the coding process. After coding, the research team engaged in iterative discussions to group related codes into preliminary themes and sub-themes. Regular meetings were held to refine definitions, merge overlapping categories, and ensure that themes were consistently represented across multiple participant responses. This process allowed researchers to continuously refine and validate emerging themes to accurately reflect participants' perspectives. To ensure accuracy, the research team reviewed, reworded, and refined the identified themes by comparing them against the original data transcripts. Disagreements regarding theme classification were resolved through team discussions. If necessary, a third researcher was consulted to decide on categorization. This step helped strengthen the internal validity of the thematic analysis. At this stage, researchers agreed upon the final theme names and definitions. The validated themes were checked against the original data to confirm that they accurately captured participant experiences. No new themes emerged at this stage, indicating data saturation.

**Reliability and trustworthiness.** To enhance the credibility and reliability of the thematic analysis, multiple validation strategies were applied:

1. Investigator Triangulation: Multiple researchers were involved in coding and analyzing the data to reduce bias and strengthen consistency.

2. Participant Validation (Reflection Session): The final themes were presented in a reflection session, where NICU nurses reviewed and confirmed their accuracy. This step ensured that the themes were authentic representations of participants' perspectives.

3. Double-Checking Method [19]: This qualitative validation technique was used to cross-check interpretations and enhance trustworthiness.

By integrating investigator triangulation, participant feedback, and data saturation techniques, this study ensured that the final themes were both theoretically grounded and empirically supported.

# Findings

All participants identified as female and reported bilingual proficiency: six in Arabic and English, two in Tagalog and English, and one in Hindi and English. Their clinical experience in neonatal intensive care ranged from under one year to nine years. Analysis revealed three overarching themes and nine associated subthemes. The findings are presented below with illustrative quotations corresponding to each subtheme.

## A. The communication challenges

Thematic analysis identified three main categories of challenges to effective communication in the NICU, as detailed in Table 1. These categories—Agency, Structure, and Culture—encompass key themes and subthemes derived from participant experiences.

**Theme one: Skills and resources for effective communication. Subtheme 1: Inadequate communication skills** Participants identified significant gaps in communication training, particularly for high-stakes scenarios such as delivering distressing news or addressing complex cases. Many nurses expressed feeling unprepared to manage these interactions, leading to anxiety and self-doubt. One participant shared:

*"... lack of skills… lack of confidence when the case is complex; most of us are not prepared for that"* (Group A).

Newly employed nurses faced heightened challenges, as parents often sought answers beyond their expertise. A nurse recalled:

*"At the beginning of my employment, I didn't know what to tell the family—what my role was versus the doctor's"* (Group A).

### Subtheme 2: Limited institutional support

Systemic barriers, including high workloads, staffing shortages, and insufficient time for parent engagement, further hindered effective communication. Participants emphasized the physical and emotional toll of balancing clinical duties with communication demands. One nurse explained:

*"With limited teamwork and [staff] shortages… we need help to reduce the pressure on us"* (Group B).

Another highlighted the strain of multitasking:

*"Every nurse cares for three babies and communicates with three mothers. Each mother deserves education and time—their right—but it's physically and emotionally draining"* (Group B).

**Table 1. Themes and subthemes of challenges.**

| Category | Themes | Subthemes |
|---|---|---|
| Agency | Skills and Resources for Effective Communication | ▪ Inadequate communication skills<br>▪ Limited institutional support |
| Structure | Communication Policies and Standards for Practice Improvement | ▪ Absence of standardized protocols |
| | | ▪ Unregulated family communication (e.g., frequent calls) |
| | | ▪ Inconsistent information sharing |
| | | ▪ Disrespectful interactions |
| Culture | Shared Understanding Amid Diversity | ▪ Language barriers |
| | | ▪ Variability in parental health literacy |
| | | ▪ Socioeconomic disparities |

**Theme two: Communication policies and standards for practice improvement.  Subtheme 1: Absence of standardized protocols**

Participants emphasized the lack of clear guidelines for communication, leading to inconsistencies in roles, timing, and content. For example, one nurse shared:

*"There's no policy for communicating with parents—not for nurses or doctors. One day, a mother gets detailed explanations; the next day, she gets only bullet points. Parents get confused and blame us"* (Group B).

This ambiguity left nurses vulnerable to parental dissatisfaction and undermined trust.

**Subtheme 2: Unregulated family communication**

Unstructured visitation hours and unrestricted phone calls disrupted workflows and compromised care quality. A participant explained:

*"Family members call hourly, even during emergencies. With staffing shortages, constant interruptions risk patient safety—babies could be left alone for critical minutes"* (Group A).

Nurses advocated for policies to manage communication channels, such as scheduled update times or dedicated family liaisons.

**Subtheme 3: Inconsistent information sharing**

Discrepancies in messaging among healthcare providers eroded parental trust. A nurse recounted:

*"A colleague told parents their baby was stable, but the infant died shortly after. The family arrived to a devastating scene—it shattered their trust in us"* (Group A).

Participants stressed the need for unified communication protocols to ensure alignment across shifts and roles.

**Subtheme 4: Disrespectful interactions**

Parents' dismissal of nurses' expertise and insistence on physician validation strained relationships. One participant noted:

*"We spend hours explaining treatments, but parents still demand to 'hear it from the doctor.' When we can't reach busy physicians, some yell at us… It's demoralizing"* (Group A).

Nurses highlighted the need for institutional support to establish professional boundaries and mutual respect.

**Theme three: Shared understanding amid diversity.  Subtheme 1: Language barriers**

Language differences emerged as a critical obstacle, particularly for non-Arabic-speaking nurses. Participants described challenges in ensuring parental comprehension and the added time required to bridge linguistic gaps. One nurse explained:

*"Language is an obstacle for me. Since I'm not an Arabic speaker, I often ask colleagues to clarify medical instructions for parents"* (Group A).

**Another noted the preference for Arabic-speaking staff**

*"Parents favor Arabic-speaking nurses—it's easier for them. When I explain discharge plans, I need an Arabic nurse to translate, which takes extra effort"* (Group B).

**Subtheme 2: Variability in parental health literacy**

Disparities in health literacy further complicated communication, as parents' understanding of medical terms and care plans varied widely. A participant shared:

*"No matter how much I explain, some parents struggle to grasp instructions. They cling to their beliefs, like refusing medications they don't understand"* (Group A).

Nurses adapted by using visual aids, as one described:

*"When words fail, I use pictures or diagrams to explain treatments. It helps, but it's time-consuming"* (Group B).

**Subtheme 3: Socioeconomic disparities**

Socioeconomic status influenced how families engaged with care teams. Participants observed that parents from marginalized backgrounds often felt dismissed or distrusted medical advice.
A nurse remarked:

*"Families' social status affects how seriously they take our guidance. Some assume we're judging them or withholding care"* (Group B).

### B. Communication infrastructure model

During the final session of the WBPD, a communication infrastructure model was developed to enhance interactions between nurses and parents in the NICU, enabling effective engagement with health information (Fig 2). The model recognizes that communication is a multifaceted process influenced by three key areas: skills and resources, policies and standards, and shared understanding. The model fosters a more structured and effective communication environment by addressing these interrelated components. The model underscores the importance of developing nurses' skills and resources through structured training and mentorship programs to strengthen communication. By equipping nurses with scenario-based communication training and guided role navigation, they gain confidence and competence in handling parent interactions. Additionally, workload reforms to optimize nurse-to-patient ratios ensure that nurses have sufficient time to engage meaningfully with families, promoting better communication outcomes. Beyond individual skill development, policies and standards are crucial in structuring communication in the NICU. The model emphasizes the need for standardized protocols that define clear roles, communication timing, and message consistency. Family communication policies, such as regulated call hours and designated liaisons, help streamline interactions, minimizing disruptions while ensuring that parents receive timely and accurate information. Furthermore, interprofessional collaboration fosters coherence in messaging by promoting shared documentation and coordinated briefings. Establishing a culture of respectful engagement through boundary-setting initiatives ensures that both nurses and parents feel supported in their interactions. The model also highlights the importance of shared understanding in fostering effective communication. Language barriers, varying literacy levels, and cultural differences can all impact how health information is received and processed. To bridge these gaps, communication strategies should incorporate language support tools, including translation apps, bilingual staff, and visual aids that facilitate clearer interactions. Additionally, health literacy assessments help gauge parents' comprehension of medical information, allowing for adapting communication styles, such as using simplified language and demonstrations. Recognizing the diverse backgrounds of NICU families, culturally sensitive training equips nurses with the knowledge and strategies needed to build trust and rapport with parents from different socioeconomic and cultural contexts. By integrating these three interconnected dimensions, the communication infrastructure model provides a comprehensive framework for overcoming communication challenges in the NICU. Through targeted training, clear policies,

and an inclusive approach, it fosters an environment where nurses and parents can engage in meaningful, effective, and culturally competent communication, ultimately enhancing the quality of neonatal care.

## Discussion

The WBPD used in this study allowed for a comprehensive exploration of communication challenges in the NICU from the nurses' perspective, utilizing the CCA to address key barriers and propose practical solutions. The findings reinforced that nurses' communication skills, institutional communication policies, and shared understanding of diversity are critical factors influencing effective communication with parents in the NICU. These elements, when addressed holistically, these elements contribute to a more structured and effective communication infrastructure.

A central theme emerging from this study is the need for structured communication training to equip NICU nurses with the confidence and competence to handle emotionally charged interactions. Participants reported gaps in preparedness when discussing complex cases or delivering distressing news, reinforcing previous findings by Bry et al. [20], who highlighted the necessity of tailored training for improving communicative competence. However, unlike prior studies that focused narrowly on individual skill-building [21], this study emphasizes a multifaceted approach that integrates mentorship, workload reforms, and interactive training to address communication challenges holistically.

Structured scenario-based workshops and mentorship programs emerged as key strategies to enhance nurses' preparedness for challenging interactions. By pairing new nurses with experienced mentors, hospitals can facilitate smoother role navigation and ensure that nurses receive guidance on effectively communicating with parents. Additionally, reducing nurse-to-patient ratios through staffing and workload reforms allows for more meaningful engagement with families, ensuring that communication is not rushed or overlooked in high-pressure environments. These elements align with Bordelon et al. [22] evidence on the efficacy of simulation-based learning in neonatal care, reinforcing the importance of interactive training approaches. This participatory approach, grounded in the CCA, shifts the focus from passive skill acquisition to

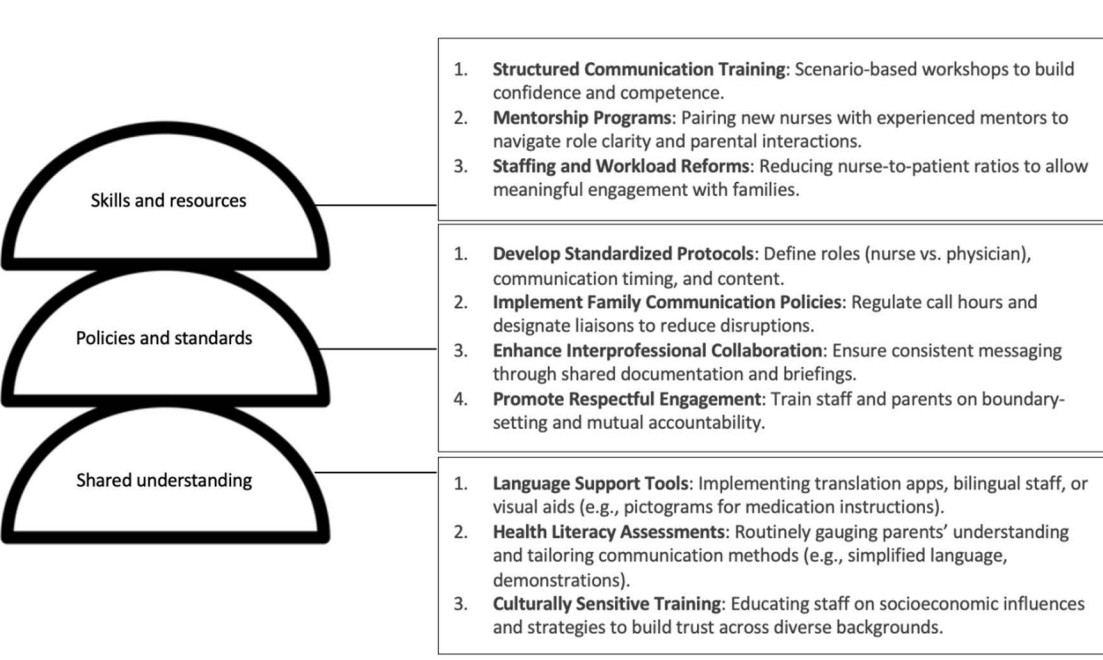

**Fig 2. A communication infrastructure model.**

co-designing training that reflects nurses' real-world needs. By incorporating structured training, mentorship, and workload considerations, this model offers a systemic rather than individualized solution to improving NICU communication, representing a novel contribution to the existing literature on nurse training and development.

The absence of standardized communication protocols emerged as a significant structural barrier, leading to parental confusion and dissatisfaction. Participants highlighted inconsistencies in how information was conveyed, with variations in detail and messaging across different nurses, ultimately undermining parental trust. This finding aligns with Biasini et al. [23], who identified similar inconsistencies in NICU communication practices. However, this study extends their work by advocating for structured communication policies incorporating standardized protocols, interprofessional collaboration, and respectful engagement to ensure clarity and trust in nurse-parent interactions. Developing standardized protocols for update timing—such as scheduling parental updates after medical rounds—and structured content—such as utilizing pre-approved templates—can enhance communication consistency while minimizing interruptions to nurses' workflow. Additionally, implementing family communication policies, including designated liaison roles and regulated call hours, ensures that families receive timely and clear information while reducing unnecessary disruptions. Strengthening interprofessional collaboration through shared documentation and cross-disciplinary briefings further enhances message consistency, reinforcing a unified communication strategy across NICU teams.

Moreover, fostering respectful engagement through targeted staff and parent training encourages boundary-setting and mutual accountability, ensuring that communication is both professional and empathetic. These policy recommendations align with Willems et al. (2021) [6] call for standardized guidelines but uniquely emphasize nurse-led input in policy development. By prioritizing practical, culturally responsive solutions co-developed with frontline staff, this model offers a structured framework for improving communication transparency and building parental trust in the NICU setting.

Cultural and linguistic barriers were identified as significant obstacles to effective communication, particularly for non-Arabic-speaking nurses. Participants reported frequently relying on colleagues or improvised visual aids to bridge these gaps, a challenge consistent with Gerchow et al.[24] global review of nurse-parent language barriers. However, this study extends the discussion beyond traditional interpreter-based solutions by emphasizing multifaceted strategies, including language support tools, health literacy assessments, and culturally sensitive training, to create a more inclusive communication framework.

Investing in language support tools, such as translation apps, bilingual staff, and pictorial discharge guides, empowers nurses to independently address language barriers, reducing dependency on external interpreters. Additionally, health literacy assessments help tailor communication methods to parents' comprehension levels, ensuring that critical medical information is effectively conveyed. Providing culturally sensitive training further equips nurses with the skills to navigate diverse interactions, fostering trust and rapport with families from varied backgrounds. This approach aligns with the CCA, which advocates for grassroots, community-driven health interventions. Much like Kerala's participatory public health models [12], the integration of nurse-led, culturally responsive communication strategies ensures that solutions are both practical and contextually relevant [25]. By prioritizing a holistic, institutionally supported model, this study advances the discourse on addressing linguistic and cultural diversity in NICU settings, offering sustainable strategies for improving nurse-parent communication.

An unanticipated finding was the persistent parental preference for physician communication, even when nurses provided detailed explanations. This reflects deeper systemic hierarchies, where nurses' expertise is undervalued—a dynamic also observed in Sisk et al. [26] pediatric oncology study. However, this study diverges by linking this issue to structural power imbalances and the lack of standardized communication protocols, a perspective rarely applied in NICU communication research. The CCA framework underscores how top-down healthcare structures marginalize nurses' voices, paralleling critiques of pandemic responses that sidelined community input [12]. To address this, fostering interprofessional collaboration, such as joint nurse-physician parent meetings and shared communication protocols, could enhance nurses' roles and authority in information-sharing, a strategy supported by Dutta et al. [12] work on team-based communication.

Similarly, while workload and staffing shortages are well-documented in nursing literature [27], participants in this study highlighted their direct impact on communication quality and parental engagement. Nurses reported struggling to provide individualized updates due to time constraints, reinforcing findings from Labrie et al. [8] meta-synthesis, which linked workload to communication lapses. However, this study advances the discourse by proposing nurse-driven mitigations, including hiring dedicated communication liaisons, redistributing non-clinical tasks, and integrating structured workload policies to ensure nurses can dedicate sufficient time to family interactions. These findings demonstrate how participatory research can translate systemic critiques into tangible policy changes, aligning with institutional reforms identified in the communication infrastructure model. By addressing both hierarchical power imbalances and workload constraints, these insights contribute to a more equitable and sustainable communication framework in NICU settings.

The findings collectively advocate for a multilevel communication infrastructure that integrates skills and resources, policies and standards, and shared understanding to create a sustainable and inclusive NICU communication framework. While Wilkin [28] Communication Infrastructure Theory informed the study's design, this work extends its application by centering nurses as co-architects of solutions, a departure from traditional top-down models. Instead of viewing communication as an externally imposed process, this model integrates structured training, institutional policies, and culturally responsive tools, ensuring that communication strategies are nurse-led, context-specific, and practical. For example, the proposed "communication bundle"—which combines structured communication training, mentorship programs, standardized protocols, language support tools, and team culture initiatives—mirrors successful interventions in pediatric ICUs [26]. However, this study innovates by embedding nurses' insights into every tier of communication reform, ensuring that proposed strategies align with frontline experiences and institutional realities. This approach aligns with the CCA and its transformative potential, as demonstrated in Māori-led health initiatives that prioritize indigenous knowledge and community-driven solutions in policy design [12].

By moving beyond hierarchical, top-down communication reforms, this study contributes a nurse-driven, participatory model that advances both theoretical discourse and practical implementation in NICU settings. The findings suggest that creating a sustainable, structured communication infrastructure—one that balances individual, systemic, and cultural considerations—can empower nurses, improve parent trust, and enhance overall healthcare delivery.

## Limitation

The study findings should be used with caution due to some limitations. First, while the sample was fairly heterogeneous and data saturation was met, there were no male participants. Gender differences in healthcare provider-parent communication should be considered in research and in-service training of nurses [29]. It is possible that additional qualitative studies considering both genders would have produced additional themes. Second, it should also be noted that the NICU nurses enthusiastically agreed to participate, thus provoking motivation bias. Third, all participants were from the same geographical locations and healthcare settings with different social backgrounds and language differences. Additional studies beyond Saudi borders could provide additional insights.

## Conclusion

This study provides a framework for building sustainable communication infrastructures by centering nurses as key stakeholders in addressing communication challenges within the NICU. Appropriate, multilevel communication infrastructures ensure that NICU nurses can effectively manage health communication demands. Traditional approaches focusing solely on individual skill-building or structural policies often overlook the interplay between personal, institutional, and cultural factors in shaping communication practices. The current study advances this understanding by utilizing the CCA to develop a multilevel intervention model. This study offers a comprehensive strategy that bridges gaps between individual capacity, systemic policies, and shared understanding by integrating structured communication training, institutional policies, and culturally responsive tools. The findings reinforce the need for nurse-led, participatory communication reforms

to foster trust, consistency, and inclusivity in NICU interactions. Investing in holistic communication infrastructures can empower nurses, enhance parental engagement, and improve overall healthcare delivery in neonatal care settings.

## Acknowledgments

The authors would like to thank the nurses who participated in this study.

## Author contributions

**Conceptualization:** Modi Al-Moteri, Raneem Mohammed Althobiti, Rahaf Talal Alsalmi, Nadeen Ibraheem Almalki., Waad Moala Alsufyani, Ghida Abdullah Alghamdi, Shahad Fahad Albogami, Mjd Fahid Alotaibe, Nora Naeem Alameen, Jamil Aljuaid.

**Formal analysis:** Jamil Aljuaid.

**Funding acquisition:** Modi Al-Moteri.

**Methodology:** Modi Al-Moteri, Raneem Mohammed Althobiti, Rahaf Talal Alsalmi, Nadeen Ibraheem Almalki., Waad Moala Alsufyani, Ghida Abdullah Alghamdi, Shahad Fahad Albogami, Mjd Fahid Alotaibe., Nora Naeem Alameen.

**Supervision:** Modi Al-Moteri.

**Writing – original draft:** Modi Al-Moteri.

**Writing – review & editing:** Maaidah M. Algamdi, Saud Owaydh Almutiry.

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
