## [Decision Letter · Decision Letter 0]

24 Feb 2025

PONE-D-24-34089Workshop-based participatory research designthrough the lens of a culture-centered approach (CCA)PLOS ONE

Dear Dr. Al-Moteri,

Thank you for submitting your manuscript to PLOS ONE. After careful consideration, we feel that it has merit but does not fully meet PLOS ONE’s publication criteria as it currently stands. Therefore, we invite you to submit a revised version of the manuscript that addresses the points raised during the review process.

We look forward to receiving your revised manuscript.

Kind regards,

Mostafa Shaban

Academic Editor

PLOS ONE

[The authors would like to acknowledge Taif University for supporting this study through the University Research Supporting Project number (TURSP-2020/194). The authors wish also to express their grateful to the nurses who participated in this study.]

 [The author(s) received no specific funding for this work.]

4. Please amend the manuscript submission data (via Edit Submission) to include author Jamil Aljuaid, Raneem Mohammed Althobiti, Rahaf Talal Alsalmi, Nadeen Ibraheem Almalki, Waad Moala Alsofyani , Ghida Abduallah Alghmdi, Shahdad Fahad Albogami, Mjd Fahid Alotaibe,  Nora Naeem Alameen, Maaidah M. Algamdi and Saud Owaydh Almutiry.

Reviewers' comments:

Reviewer's Responses to Questions

**Comments to the Author**

1. Is the manuscript technically sound, and do the data support the conclusions?

Reviewer #1: Yes

2. Has the statistical analysis been performed appropriately and rigorously? 

Reviewer #1: N/A

3. Have the authors made all data underlying the findings in their manuscript fully available?

Reviewer #1: Yes

4. Is the manuscript presented in an intelligible fashion and written in standard English?

Reviewer #1: Yes

5. Review Comments to the Author

Reviewer #1: Thank you for sharing your manuscript. It tackles an important and relevant topic—improving NICU communication through the voices of nurses, which is a valuable contribution to the field. The use of a participatory approach and the focus on cultural sensitivity through CCA are both commendable.

• It’s unclear how this study differs from previous research. Be more explicit about how this study fills the gap in the existing literature.

• The role of researchers is vague. Clarify what exactly the researchers did—were they facilitators, analysts, or something else?

• The mention of the Culture-Centered Approach (CCA) is good, but it would be helpful to briefly explain what CCA is for readers unfamiliar with the term.

• The three themes are clear but lack some depth. How were these themes derived? Was there a specific process (e.g., coding, group discussions)?

• The conclusion is too vague. The statement about "producing significant changes" needs more detail—what exactly will change in NICU communication?

• The abstract could mention the broader implications of the study. How can these findings be applied beyond just the NICU setting?

• The abstract needs more detail about the impact of the findings. Be specific about how the proposed changes in communication infrastructure will improve NICU practices.

Introduction

• The introduction could provide more context about the existing literature on health communication in the NICU. It's important to clearly establish why the focus on nurses’ voices is necessary and how it adds to current research.

• The introduction could benefit from a more concise description of the research problem. Some sections are a bit repetitive, especially when talking about the importance of nurses’ involvement. Streamlining this could make the introduction more focused.

• The introduction should include a clear research question or hypothesis. While the purpose is mentioned, framing it as a specific research question or problem to address will help readers follow the study’s goals more clearly.

• The role of health communication in improving outcomes for both parents and newborns is mentioned, but it could be expanded. How does improving communication between nurses and parents directly affect patient outcomes or experiences in the NICU?

• The introduction might be improved by framing the study in a broader context. How does this research contribute to the wider field of health communication or nursing research? A few sentences on the broader impact could strengthen the introduction.

Methodology

• The role of nurses in the workshops isn’t clearly defined—were they participants, co-facilitators, or something else?

• The structure of the workshops needs more explanation— were there, and what was the format?

• The process of thematic analysis needs more detail. How were the themes identified, and was there a process for ensuring reliability in the analysis?

• Overall, the methodology seems appropriate, but more detail is needed to make it clearer, especially regarding the roles of participants, data collection, and ethical safeguards.

Procedure

• The procedure is not detailed enough in terms of how the workshops were conducted. For example, how were participants recruited and how were the workshops organized?

• There’s no mention of how long each workshop lasted or how many total sessions were held. This is important to assess the depth and quality of the participatory process.

• Facilitation process is unclear. Were the workshops led by the researchers, or were NICU nurses involved in facilitating some of the discussions? Clarify the role of the researchers and the nurses in the procedure.

• It’s not clear if there was any follow-up with participants after the workshops. Did researchers engage with the nurses again to discuss the impact of the workshops or to follow up on outcomes?

• The method of data collection (e.g., observations, notes, audio recordings) is not described. It’s important to explain how the data was gathered during the workshops and whether any tools were used (e.g., surveys or interview guides).

Results and discussion

• The themes (nurses’ skills, communication policies, and shared understanding of diversity) are relevant, but the discussion could benefit from more exploration of how these themes specifically impact NICU practices or improve communication with parents.

• There’s no discussion of any conflicting data or unexpected results. If there were any surprising findings or issues that emerged, it would be valuable to mention them and discuss how they were addressed.

• The implications of the findings are mentioned, but the discussion could explore them in more detail. How exactly can these findings be applied to improve health communication in the NICU? What are the practical steps forward based on these results?

• There’s a need for a more detailed comparison between the results of this study and previous research. What does this study contribute that’s new or different? How do the findings compare to other studies in the NICU or health communication field?

6. PLOS authors have the option to publish the peer review history of their article (what does this mean? ). If published, this will include your full peer review and any attached files.

**Do you want your identity to be public for this peer review?** For information about this choice, including consent withdrawal, please see our Privacy Policy .

Reviewer #1: No

---

## [Editor Report · Decision Letter 1]

28 Mar 2025

Workshop-based participatory research designthrough the lens of a culture-centered approach (CCA)

PONE-D-24-34089R1

Dear Dr. Al-Moteri,

We’re pleased to inform you that your manuscript has been judged scientifically suitable for publication and will be formally accepted for publication once it meets all outstanding technical requirements.

Kind regards,

Mostafa Shaban

Academic Editor

PLOS ONE
---

## [Editor Report · Acceptance letter]

PONE-D-24-34089R1

PLOS ONE

Dear Dr. Al-Moteri,

I'm pleased to inform you that your manuscript has been deemed suitable for publication in PLOS ONE. Congratulations! Your manuscript is now being handed over to our production team.

Kind regards,

on behalf of

Dr. Mostafa Shaban

Academic Editor

PLOS ONE